# Health and sustainability of glaciers in High Mountain Asia

Evan Miles [1✉], Michael McCarthy [1,2], Amaury Dehecq [1,3], Marin Kneib [1,4], Stefan Fugger [1,4] & Francesca Pellicciotti [1,5]

Glaciers in High Mountain Asia generate meltwater that supports the water needs of 250 million people, but current knowledge of annual accumulation and ablation is limited to sparse field measurements biased in location and glacier size. Here, we present altitudinally-resolved specific mass balances (surface, internal, and basal combined) for 5527 glaciers in High Mountain Asia for 2000–2016, derived by correcting observed glacier thinning patterns for mass redistribution due to ice flow. We find that 41% of glaciers accumulated mass over less than 20% of their area, and only 60% ± 10% of regional annual ablation was compensated by accumulation. Even without 21st century warming, 21% ± 1% of ice volume will be lost by 2100 due to current climatic-geometric imbalance, representing a reduction in glacier ablation into rivers of 28% ± 1%. The ablation of glaciers in the Himalayas and Tien Shan was mostly unsustainable and ice volume in these regions will reduce by at least 30% by 2100. The most important and vulnerable glacier-fed river basins (Amu Darya, Indus, Syr Darya, Tarim Interior) were supplied with >50% sustainable glacier ablation but will see long-term reductions in ice mass and glacier meltwater supply regardless of the Karakoram Anomaly.

[1] Swiss Federal Institute for Forest, Snow and Landscape Research WSL, Birmensdorf, Switzerland. [2] British Antarctic Survey, Natural Environment Research Council, Cambridge, UK. [3] Laboratory of Hydraulics, Hydrology and Glaciology, ETH Zurich, Zurich, Switzerland. [4] Institute of Environmental Engineering, ETH Zurich, Zurich, Switzerland. [5] Department of Geography, Northumbria University, Newcastle, UK. ✉email: evan.miles@wsl.ch

Glaciers and snow in High Mountain Asia (HMA) release enough meltwater seasonally to meet the requirements of nearly a quarter of a billion people[1,2], and basins fed by these glaciers are the most vulnerable worldwide to ongoing climatic, societal and environmental change[3]. Assessing the current state and future prevalence of ice and snow reservoirs in HMA is therefore a key priority[4–6]. However, access to the remote, high-altitude glaciers of HMA can be dangerous and time-consuming, which has restricted field observations of surface mass balance to a sparse coverage of very few sites, mostly at lower-altitude ablation areas[7,8]. Mass inputs to glaciers are generally unknown, due to the high uncertainty of reanalysis data at high altitudes and few direct observations[9–11]. The current models of glacier change are thus overparameterized and unable to constrain key aspects of glaciers' internal dynamics and interactions with climate, such as the influences of avalanching and debris cover[12]. The observational uncertainty in glacier state and varying process representation thus leads to considerable uncertainty in glacier volume change projections within HMA and globally[6,13].

Recent remote sensing studies have advanced regional-scale understanding of the mass change and dynamics of HMA glaciers[14–20]. However, even high-precision measurements of elevation change cannot resolve the spatial patterns of mass balance across individual glaciers[21]. This specific mass balance (SMB), sometimes called the climatic mass balance, is the combination of accumulation (mass gain) and ablation (mass loss) at a position on the glacier, and combines surface, englacial, frontal, and basal components. Local mass gains and losses due to accumulation and ablation are partially compensated by mass redistribution as a result of ice flow, leading to ice thinning (negative elevation change) or thickening (positive elevation change, Supplementary Figs. 1-2); rather than representing the local signals of SMB, elevation change measurements integrate ice motion[22,23]. In addition, average glacier mass balances estimated from thinning datasets alone typically use an area-averaged glacier surface density which may not be appropriate for glaciers severely out of balance with climate[24].

Knowledge of SMB is vital for understanding the regional and local drivers of glacier change[7,25], and for calibration and validation of numerical models to appropriately represent current and future glacier changes[12,13,26]. As the basal and englacial components of SMB are often negligible, while frontal ablation is localized, observations and models often equate SMB with the surface mass balance. New generations of glacier models have incorporated improved spatial representations of glacier surface processes and ice dynamics[13,27], but without spatially resolved control datasets, these models are forced to calibrate to sparse, biased measurements of surface mass balance or spatially integrated signals of glacier change such as thinning or area change[5,12,28]. The scarcity of in situ measurements is particularly problematic in HMA because many of the region's glaciers do not conform to the mass balance patterns assumed by regional-scale models. SMB is typically simplified to linear altitudinal gradients for the ablation and accumulation areas but the prevalence of avalanching[29,30] and supraglacial rocky debris[5,31] across the region may lead to distinctive mass balance profiles[32], while mass accumulation rates above 6000 m a.s.l. are rarely measured[9,33–35].

In this study, we provide spatially distributed SMB for glaciers across the entirety of HMA for the 2000–2016 period. We use this dataset to derive glacier mean equilibrium line altitudes (ELAs), the extent of accumulation areas, and the portion of annual ablation that is compensated by annual accumulation as indicators of glacier health in major river basins across the region. Finally, we assess the consequences of the glaciers' contemporary climatic-geometric imbalance in terms of implied changes in ice volume and discharge by 2100.

## Results and discussion

**Altitudinal SMB**. We determine SMB by solving the continuity equation, assuming that englacial and basal mass change is negligible (Eq. 1, "Methods"). We calculate the ice flux divergence by combining estimates of ice thickness[36] with observations of ice surface motion[16] and a Monte Carlo-based estimate of depth-averaged correction factor. We use this with elevation change measurements[15] to derive fully distributed and altitudinal SMB, carefully accounting for uncertainty associated with the input data and methods (Methods). Our results correspond to mean annual values for the 2000–2016 period, as constrained by the input elevation change and velocity observations.

The continuity equation has previously been used to determine the SMB on individual glaciers[22,23,37–40] but never at a large scale, which requires an automated approach. We first apply our approach to 35 sites to compare to available measurements of surface mass balance[41] (Supplementary Table 1), and another 25 glaciers for which remote sensing-derived estimates of SMB are available[38]. Our method is consistent with 79% of field measurements to within 0.2 m w.e. a$^{-1}$ and generally reproduces observed mass balance patterns where glacier velocity is measurable (Supplementary Information). We thus apply this method to the 7341 glaciers in High Mountain Asia with all necessary inputs and an area of 2 km$^2$ or greater, for which velocity is generally resolved well. We remove those that are known to be surge-type[42], and glaciers with inverted or erratic elevation change or mass balance profiles (Supplementary Information), indicative of erroneous input data or undocumented surge behavior, which is common for larger glaciers in this region[43]. The final set of 5527 glaciers represent 71% of the total volume (56% of total area) of glaciers larger than 2 km$^2$ in the region.

We present the area-weighted mean profile of SMB relative to each glacier's elevation range for all of HMA in Fig. 1, as well as each glacier's mean SMB resolved from our method. The difference between SMB and thinning patterns (Fig. 1b) strongly underlines the necessity of accounting for ice flow[23], and enables our method to resolve accumulation and ablation areas. By representing density differences in accumulation and ablation areas (Methods), our results reveal a consistent bias of +0.07 m w. e. a$^{-1}$ in past estimates relying on a single density value (Fig. 1, Supplementary Fig. 11). We calculate more negative mass balances despite our subset of glaciers exhibiting a slight positive bias in terms of volume change, and this bias exceeds the reported uncertainty in many subregions (Table S2).

The subregional SMB profiles (Supplementary Figs. 11, 12) emphasize differences in subregional glacier health. Consistent, distinct mass balance gradients are evident for accumulation and ablation areas. Also apparent for many subregions is a mass balance gradient reversal in the lowest elevations, attributable to inverted SMB gradients in debris-covered areas[32,38] and erroneous input data (Fig. S31). The Nyainqentangla subregion shows the most negative SMB profile, with glaciers accumulating mass over only the upper 20% of their elevation range. The Everest, Spiti Lahaul, and Tien Shan subregions exhibit similar normalized elevation SMB profiles despite occupying very different ranges. Glaciers in these subregions accumulate mass over the upper 40% of their elevation range. In contrast, glaciers in the approximately neutral-balance Kunlun Shan and Karakoram accumulate mass over the upper 60% of their elevation range and exhibit less negative SMB in ablation areas.

**ELAs and accumulation area ratios (AARs)**. We leverage the distributed SMBs to resolve ELAs and AARs for each glacier (Fig. 2, "Methods"). The area-weighted ELA for the entire region

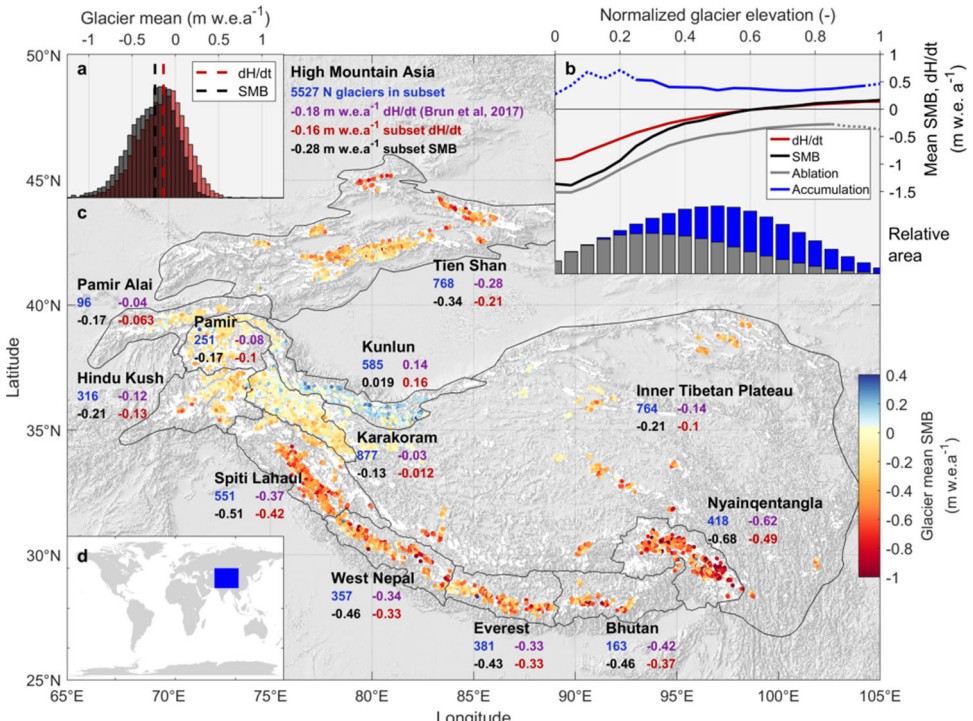

**Fig. 1 Summary of the altitudinally resolved specific mass balance (SMB) results of this study. a** Mass balance of glaciers in High Mountain Asia based on our specific mass balance results and compared to that derived from elevation change data (d$H$/d$t$)[15]. **b** Regional area-weighted mean SMB and d$H$/d$t$ for the 2000–2016 period, also indicating mean rates of accumulation and ablation, all shown with elevation normalized to each glacier's elevation range. **c** The spatial pattern of glaciers analyzed and our results for glacier and subregional mass balances (SMB), also compared to the d$H$/d$t$ results for each subregion and the glacier subset that we process. Uncertainty and subregional profiles are shown in Supplementary Figs. 8, 9. Glaciers we did not process are shown in white, and the background is hillshade of the GTOPO30 dataset sourced from the USGS (https://doi.org/10.5066/F7DF6PQS). **d** Geographic position of the HMA domain.

is 5283 m a.s.l., and the regional AAR is 0.51 (Table S3). Our glacier-specific ELA results extend the local perspective of previous studies to the entirety of the region. Studies of individual glaciers in the Tibetan Plateau and in the Central Himalaya have shown ablation areas extending to 6000 m a.s.l.[33,44], but our results show that this is true for many glaciers on both sides of the Himalayan Arc and throughout western Tibet (Fig. 2). This is not due to a bias in our glacier-specific ELA values, as they agree with the few reported ELAs based on seasonal snowline elevations and debris extent (Figs. S83–85). Crucially, our ELA results provide a unique dataset that can enable a novel calibration and validation of glacier models[12,45].

ELAs and AARs vary considerably between glaciers across the region, with standard deviations of 678 m and 0.32, respectively. ELAs follow broad topographic variations, as glacier extent is limited by the intersection of climate with topographic availability. However, we find that the median glacier elevation (a commonly used proxy for ELA) and ELA clearly differ (median absolute deviation of 193 m, Fig. S16), emphasizing the importance of accounting for each glacier's geographic and climatic setting[10]. In the Nyainqentanglha subregion, for example, many glaciers exhibit low ELAs (Fig. 2a) despite losing mass rapidly[14,15,20]; these maritime glaciers are sustained by high annual precipitation[46], but are highly sensitive to warming[45].

The AAR indicates the portion of glacier area gaining mass at the surface, and thus reflects the glacier's relative health within its local context. AARs are typically in the range of 0.5 to 0.8 for mountain glaciers roughly in climatic-geometric balance, and are lower for glaciers losing mass[32]. Numerous glaciers in the Karakoram and Kunlun Shan had large accumulation areas (AAR > 0.5) in the early 21st century, a clear manifestation of the

'Karakoram Anomaly'[47]. Our results show that some glaciers in the neighboring Pamir, Pamir Alai, and Tibetan Plateau subregions also have high AARs (Fig. 2b). Across HMA, 40% of glaciers have AAR > 0.5, but such glaciers are very uncommon beyond the area of the Karakoram Anomaly (Fig. S17). Strikingly, the influence of the Karakoram Anomaly is not discernible in the spatial distribution of ELAs; neither the Karakoram nor the Kunlun Shan shows lower ELAs than adjacent subregions (Table S3). The smooth variations of ELA but the abrupt change of AAR around Karakoram Anomaly glaciers suggests that topographic factors might contribute to the currently stable regional mass balance. That is, the glaciers within this zone may be exceptional in part because there is an extensive high-elevation area available for them to accumulate snow, unlike in other regions (Fig. S19)[48,49]. Consequently, recent increases in high-altitude precipitation[50] would affect a disproportionately large glacier area in this subregion.

Contrasting with the high AARs of the Karakoram Anomaly glaciers, for 16% of studied glaciers (10% of area) no accumulation area exists and annual net loss occurs at the surface across all elevations (Fig. 2b). 32% of glaciers have very small accumulation areas (AAR < 0.1), accounting for 19% of glacier area, while 41% (23% of area) have AAR < 0.2. Smaller glaciers exhibit lower AARs in our results (Fig. S18), and some of these glaciers may be cases where the observed velocity is erroneously low. However, our comparison to reference measurements (Supplementary Information) highlights numerous small, nearly stagnant glaciers where the observed surface lowering corresponds directly to SMB measurements, suggesting that mass replenishment due to ice flow is negligible. Considering only the 1982 glaciers larger than 5 km², which are more likely to have measurable surface

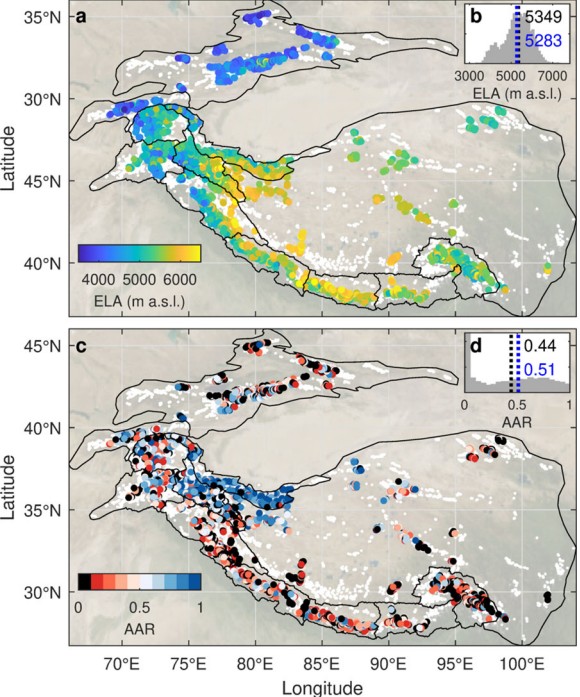

**Fig. 2 The state of studied glaciers in High Mountain Asia in 2000–2016.**
**a** The spatial pattern of equilibrium line altitudes (ELAs) determined in this study. White dots depict the full distribution of glaciers in the region. Black lines depict the outlines of subregions following[14]. **b** The region-wide histogram of ELA values, also indicating median (black) and area-weighted mean (blue) values. **c** The spatial pattern of accumulation area ratio (AAR) values determined in this study. **d** The region-wide histogram of AAR values, also indicating median (black) and area-weighted mean (blue) values. Background is the August 2004 NASA Blue Marble[75].

velocity[16], we still find that 7.5% of glaciers have an AAR < 0.1. Low-AAR glaciers are most concentrated in Eastern Nyainqen-tanglha, which is the subregion with the highest rates of mass loss. Here, 50% of glaciers have an AAR less than 0.2 (Supplementary Table 3, Supplementary Fig. 18). Such "headless" glaciers are also surprisingly common across the rest of the Himalaya, Tibetan Plateau, and Tien Shan (40% of glaciers in these regions); are less frequent in the Pamirs, Hindu Kush, and Karakoram (30%); and are rare in the Kunlun Shan (8%).

Our AAR results depict a picture of strong imbalance for most glaciers in HMA. The very low AAR values for much of HMA suggest that accumulation areas have been substantially reduced by increasing ELAs. However, we note that AARs can be depressed to 0.3–0.5 for heavily debris-covered glaciers sustained by avalanches[32], which are common in parts of HMA. Thus, although the AAR pattern highlights the contrast between glaciers affected by the Karakoram Anomaly and the rest of the region, this metric may be biased in some areas. To assess glacier health in an unbiased manner, we determine an additional indicator of glacier health: the ablation balance ratio.

**Sustainable ablation in major basins.** We use our SMB results to assess the origin of glacier ablation in the major basins draining HMA by partitioning the total annual glacier ablation into "imbalance" and "balance" components (Fig. 3, Supplementary Table 4). The "balance" component is the glacier ablation that is compensated on an annual basis by accumulation (Methods), and which we consider to be sustainable in early 21st-century conditions[1]. Crucially, the SMB results allow us to determine these values directly for each glacier, whereas prior available

estimates were obtained only at the basin scale by comparing observed thinning with glacier models[1,20]. Our results indicate that 40 ± 11% of ablation from HMA glaciers is unsustainable in early 21st century conditions. Although this indicates that the glacier contribution to streamflows will eventually reduce, there is considerable variability between regions and individual glaciers.

Basins fed by the Karakoram Anomaly glaciers (Indus, Amu Darya, Syr Darya, Tarim Interior) are the most important water towers globally[3]. The prevalence of surging glaciers in these basins leads to a relatively smaller sample size in our study (Supplementary Table 4), and these basins exhibit higher uncertainty in the absolute mass change of glaciers affected by the Karakoram Anomaly[15,20,47]. Nonetheless, we show that these basins' glaciers were much healthier compared to the rest of HMA for our study period, with over 50% of annual glacier ablation balanced by accumulation and numerous individual glaciers exceeding 100% balanced ablation (Fig. 3). The Indus basin is of particular interest due to the high dependence of its downstream populations on snow and ice melt, especially in drought conditions[1,2]; its high vulnerability to future societal and environmental change[2,3]; and the high ice-melt content of its summertime streamflows[1]. For this key basin, we find that 65% ± 23% of glacier ablation was balanced by accumulation during 2000–2016 (Supplementary Table 4).

Contrasting with basins supported by the Karakoram Anomaly glaciers, nearly all other basins' supply of glacier ablation is primarily imbalance ablation. Among these, the Ganges–Brahmaputra basin stands out as a vulnerable basin with important water supplies (Supplementary Fig. 21). Although the Ganges and Brahmaputra rivers are sustained by strong monsoonal precipitation, glaciers provide the majority of stream-flow in drought years[1]. We show that the majority of ablation from these glaciers is imbalance ablation (the balance ratio is 48% ± 9%, Fig. 3, Supplementary Table 4). The future decline in glacier meltwater supply in this region may seem relatively minor on an annual basis due to precipitation excess in normal years[12] but is crucial seasonally, affecting the growth of cash crops in the water-scarce pre-monsoon[51]. Given the likelihood of considerable cryospheric and environmental change, the societal pathway of adaptation to change in these basins will directly control downstream communities' resilience to water resource change[6,52].

A previous study[1] determined a regional balance ratio of 38% for drought years; our results indicate that a small balance ratio (40%) is the recent-period norm, rather than the exception. Our results differ at the subregional level: we find a 32% higher balance ratio for the Ganges and Brahmaputra than ref. [1], but 6–18% lower balance ratios for the basins fed by the Karakoram Anomaly (Supplementary Table 5). Our basin ablation ratios are less than those of ref. [20] by 15–25% for all basins except the Tarim, where the sign of mass balance is uncertain[15,20]. Although our imbalance ablation results are slightly higher due to our representation of density, our total ablation estimates are much lower (Supplementary Table 5). This is likely due to compensation of melt and accumulation errors in the model used by that study, which was overparameterized and did not represent the regionally important influence of debris cover[12]. Constrained by thinning rates alone, the model would overestimate total melt and compensate this error by exaggerating accumulation, leading to higher balance ratios.

The balance ratios for the Amu Darya, Tarim Interior, and Indus basins are particularly affected by the uncertainty and heterogeneity of subregional volume change, demonstrating the need for systematic measurements in this region[53], particularly of accumulation rates[34]. The sustainability of river discharge in these important and vulnerable basins is dependent on the anomalous health of glaciers in the Kunlun Shan and Karakoram ranges[47]. Despite the higher balance ratios in these basins, a

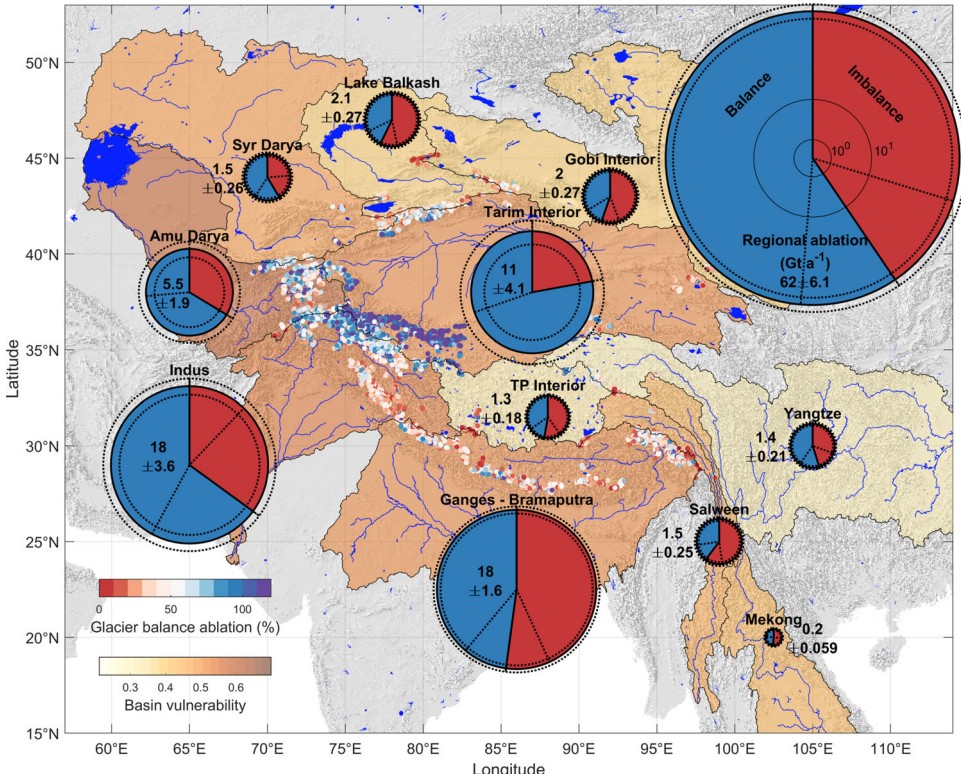

**Fig. 3 The quantity and context of glacier ablation for principal drainage basins in High Mountain Asia.** Analyzed glaciers are colored according to the portion of total annual ablation that is compensated by accumulation, which is greater than 100% for glaciers gaining mass. The portion of balance ablation derived from our results is shown for major river basins, indicating uncertainty with the dashed lines, and scaled by area according to the total estimated glacier ablation within each ("Methods", Supplementary Tables 4, 5). Basin vulnerability is colored according to the global range[3]. Background is a hillshade of the GTOPO30 dataset sourced from the USGS (https://doi.org/10.5066/F7DF6PQS).

considerable portion of glaciers are losing mass. Our results show that the major tributaries of the Indus are supplied by glaciers in contrasting health: the Chenab and Satluj are supplied by imbalance ablation from unhealthy glaciers in the Western Himalayas, while the Indus itself is supplied by imbalance ablation from the Ladakh Range and balance ablation from the Karakoram. Combining these distinct signals, our results indicate that despite the near-neutral mass balance in the Karakoram, 19% ± 12% of subregional ablation was imbalance ablation in the early 21st century (Supplementary Table 3).

**Implied glacier and ablation changes.** The considerable imbalance ablation for 2000-2016 indicates climatic-geometric disequilibrium across the region. We determine how glaciers would respond to this disequilibrium, if maintained, and find that early 21st century mass balance regimes imply a change of −23% ± 1% of glacier volume in HMA by 2100 (Fig. 4, Methods). All subregions along the Himalaya lose at least 35% of their present-day volume by 2100, contrasting with volume reductions of 10-20% for the Karakoram, Pamir Alai, Pamir, and Hindu Kush and a volume gain of 2.1% for the Kunlun Shan (Supplementary Table 4). Our results indicate that 25% of glaciers across the region would lose at least 50% of their current ice volume by 2100 without any warming (Supplementary Figs. 22, 23). We calculate a more negative long-term volume change of −34% ± 2% by 2200 than the 27–29% committed loss estimate based on field observations[25]. By resolving the implied volume change for a large population of glaciers individually, we mitigate against the biases of sparse glaciological measurements[54]. Our estimate is only slightly lower than recent projections of 29% ± 12%[12] to 36% ± 7%[5] mass loss by 2100 under the RCP2.6 climate scenario,

suggesting that under that climate pathway most glacier loss is already committed by current climatic-geometric disequilibrium.

Associated with the regional losses in glacier volume, we find a regional change of −28% ± 6% in total annual ablation rates by 2100 (Fig. 4). Subregions experience variable changes in glacier ablation largely following the changes in glacier volume, but the ablation changes are stronger than volume loss in the Karakoram and Pamir subregions. As with the volume change estimates, the projected general reduction in glacier ablation should be taken as a baseline estimate of change. Although continued 21st-century warming will lead to a peak in glacier meltwater supply[55], this will exacerbate glaciers' climatic-geometric imbalance and lead to more severe eventual reductions in glacier ablation.

We note that the mass losses are partly obscured by the Karakoram Anomaly: other than the globally unique mass gains in the Kunlun Shan and parts of the Karakoram, the region has an implied volume loss of 31% by 2100. Considering recent and further climate warming, our results represent minimum estimates of future volume loss; sustained warming would be likely to overcome recent increases in snow accumulation in the Karakoram and Kunlun Shan[47,50], exacerbating regional glacier loss. Current projections for the Karakoram and Kunlun Shan show ice losses of 10–35% by 2100 in response to continued but reduced emissions under RCP2.6, and substantial ice losses of 30–60% for RCP4.5[5,6,12]. Consequently, disentangling the causes of the Karakoram Anomaly[47] and understanding its resilience to 21st-century warming remain key priorities for scientists and stakeholders alike.

**Implications for glacier modeling and monitoring.** Our baseline estimate of 23% ± 1% volume loss by 2100 will be exceeded given that most climate trajectories indicate continued

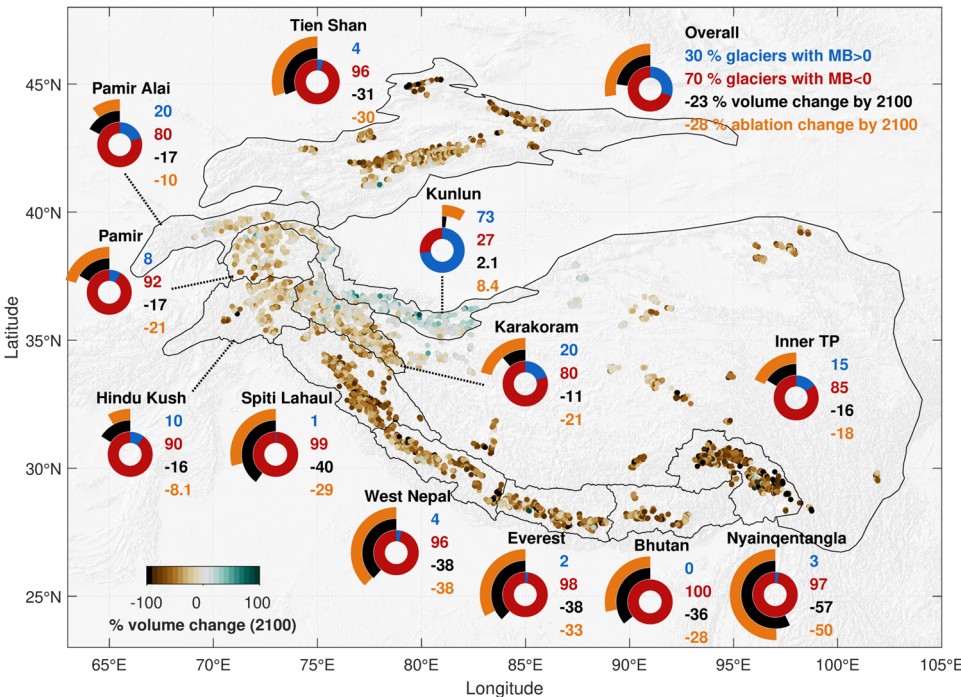

**Fig. 4 Changes in glacier volume and total ablation by 2100 implied by 2000-2016 mass balance regime, based on a glacier retreat and advance flow parameterization.** Regional icons depict the portion of glaciers with positive (blue) or negative (red) glacier-wide mass balance, implied volume change by 2100 as portion of glacier volume (black), and implied change in total annual glacier ablation by 2100 as a portion of current total annual ablation (orange). Background is a hillshade of the GTOPO30 dataset sourced from the USGS (https://doi.org/10.5066/F7DF6PQS).

warming[5,12] and the progressive deglaciation of this region will lead to a cascade of changes to ecosystems and society[6,52]. We advocate for the improvement of dynamic glacier models (e.g.[27]) to better reproduce the long-term mass balance of glaciers in HMA by including key unrepresented processes[12] such as localized mass accumulation due to avalanches, reversed mass balance gradients due to supraglacial debris, and frontal ablation due to ice-marginal lakes[17]. Spatially explicit glacier models should not be calibrated to glacier thinning datasets alone, which leads to the compensation of SMB and flux divergence errors, and can lead to errors in both melt and accumulation. Instead, models should be constrained with both glacier thinning and surface velocity observations. Our results extend the sparse glaciological measurements in HMA and thus provide the opportunity for novel strategies[12] to calibrate mass balance models directly to each glacier's altitudinally resolved SMB, or to regionally resolved mass balance gradients (Supplementary Information).

Our method has the potential to generate very novel datasets and understanding of SMB patterns worldwide, but it also demonstrates the need for improvements to existing observations. Glacier basal condition and ice rheology are poorly known at all but a few study sites. Novel field measurements of these properties would enable the uncertainties around SMB determined through a continuity approach to be significantly reduced. Robust assessments of elevation change rates are now possible at the regional scale[20], but the established average density value[24] should be reconsidered for glaciers with small accumulation areas. Problems in the input datasets of velocity and ice thickness forced us to discard results for 25% of the 7341 glaciers analyzed, and prevented application to glaciers smaller than 2 km$^2$. Ice thickness is generally the most uncertain input dataset, and additional ice thickness measurements are needed across HMA to constrain ice thickness models[56], especially for the region's debris-covered areas. New analyses of modern high repetition, high-resolution satellite data are likely to resolve flow patterns in

problematic areas such as small glaciers, tributary junctions, and icefalls[57].

Despite these challenges, we have resolved multidecadal SMB profiles across HMA, providing detail of the region's heterogeneous glacier health. We show that imbalance ablation, not replenished by annual snow accumulation, dominates the contribution of glaciers into most river basins, with the exception of basins fed by the Karakoram Anomaly glaciers. 35% of glaciers across the region are very unhealthy and are expected to lose at least half their volume by 2100 without additional climate warming. Our results provide a novel, spatially extensive dataset to calibrate and validate a new generation of glacier models capable of resolving glacier mass balance and ice dynamics at high temporal and spatial resolution. This approach paves the way to resolve the SMB across glaciers globally and for multitemporal periods to characterize the trajectory of glacier change.

## Methods

**Continuity approach to mass balance calculation.** Our mass balance reconstruction approach solves the continuity equation (Eq. 1, Supplementary Fig. 1). For any area of the glacier, in Eq. (1), d$H$/d$t$ is the annual rate of elevation change at the glacier surface, $\dot{b}$ is the annual SMB (surface, internal, and basal mass balance combined, with frontal ablation if relevant) of that area, and $\nabla*\mathbf{q}$ the annual flux divergence (determined below), accounting for the density $\rho$ for each quantity[22,23,58]. We aim to calculate $\dot{b}$, which, assuming that frontal ablation and the englacial and subglacial components are negligible, is the surface mass balance. Crucially, however, this does not equate to the glacier's melt, as it is an integrated signal for each pixel, which may experience seasonal accumulation and ablation.

$$\frac{\rho_{dH}}{\rho_{H_2O}}\frac{dH}{dt} = \dot{b} - \frac{\rho_{\nabla q}}{\rho_{H_2O}}\nabla*\mathbf{q} \tag{1}$$

We apply the continuity equation on a fully distributed basis. For this, we use ASTER-based 2000-2016 annual surface lowering trends[15], ITS_LIVE HMA ice surface velocity products[16,59] and multi-model consensus ice thickness estimates[36] which correspond to the Randolph Glacier Inventory version 6.0 outlines[42]. These datasets are available in different projections and spatial resolutions. For each glacier, we define a grid for our analysis using the local projection used by[36], and

we vary the grid resolution based on the size of the glacier: 50 m for small glaciers ($<15$ km$^2$), 100 m for larger glaciers (up to 80 km$^2$), and 200 m for very large glaciers ($>80$ km$^2$). We reproject and resample the surface lowering data (provided at 30 m resolution) and its stated uncertainty to this grid using a cubic spline. We used cubic splines to reproject both ends of the surface velocity vectors to preserve true orientation before resampling these data and their stated uncertainty from their 240 m resolution. Finally, we degrade the corresponding ice thickness data (provided at 25 m resolution); possible concerns of circular analysis with this dataset are mitigated by the method's performance for debris-covered areas and with tests using distinct individual ice thickness models (Supplementary Information). To maintain a continuous dataset over each glacier, we do not filter the surface velocity and surface lowering datasets before reprojection, but instead assess the uncertainty through our calculations.

**Ice flux and flux divergence**. We calculate the ice flux vector **q** at each cell according to Eq. (2), where $h$ is the ice thickness (m) and $\mathbf{u}_s$ is the annual ice surface velocity vector (m a$^{-1}$).

$$\mathbf{q} = h\gamma\mathbf{u}_s \qquad (2)$$

$\gamma\mathbf{u}_s$ represents the column-average ice velocity, with the constant $\gamma$ representing the relative importance of basal motion and vertical ice shear deformation (Supplementary Fig. 1). We model $\gamma$ for each glacier individually. For this, we use a Monte Carlo analysis to estimate the depth-integrated velocity at a point assuming simple shear with an assumed ratio of ice motion attributable to basal sliding $\left|\frac{u_b}{u_s}\right|$ and a thickness estimate. For a given ice thickness and basal sliding ratio, we calculate the velocity at each depth following[60], then determine $\gamma\mathbf{u}_s$. We perform this calculation for 10,000 sets of randomly drawn values of ice thickness, flow rate factor $n$, and basal sliding for each glacier. For the ice thickness distributions we use the distribution of ice thickness values produced for that glacier by[36]. For $n$ we note that $n = 3$ is appropriate for many glacier modeling situations[58] and use a Gaussian distribution with ($\mu = 3$, $\sigma = 0.067$). For the portion of flow attributable to sliding, this is dependent on both ice rheology and basal state. Neither basal sliding nor ice internal thermal profiles are well constrained for glaciers in HMA, but authors have variously assumed or determined temperate, cold, and polythermal regimes across the region[61–66], demonstrating variable thermal regimes and basal conditions across High Asia. We acknowledge that (1) many small, high-altitude glaciers are likely to be cold-bedded[64], but (2) there is increasing evidence that the lower-elevation tongues are polythermal with temperate beds[65,66]. In addition, although there are many large proglacial lakes in the region which are known to affect terminus ice velocities[17,67], it is not likely that an extensive portion of glacier ice approaches flotation. Nonetheless, without widespread knowledge of the importance of basal sliding across HMA, we assume a uniform distribution across [0,1] for our basal sliding factors. In addition to providing an estimate of $\gamma$, this Monte Carlo approach allows us to estimate its uncertainty, $\sigma_\gamma$.

The flux divergence $\nabla*\mathbf{q}$ represents the vertical component of ice velocity at the glacier surface, which leads to submergence in areas of divergent flow and emergence in areas of convergent flow. We calculate $\nabla*\mathbf{q}$ on a pixel basis using a centered-difference scheme based on the divergence of **q** (Eq. 3, Supplementary Fig. 1).

$$\nabla*\mathbf{q} = \frac{\partial q_x}{\partial x} + \frac{\partial q_y}{\partial y} \qquad (3)$$

**Density correction**. Our continuity approach assumes the mean ice density within the domain does not change with respect to time. This is generally reasonable in the ablation area or over a period that densification processes can be considered constant, leading to uncertainties on the order of 2%[58]. However, the density of snow, firn, and ice at the glacier surface must still be accounted for in order to derive $\dot{b}$. Geodetic studies often use a single value of 850 kg m$^{-3}$ or zonal values for accumulation and ablation areas[24].

We first assume that all ice fluxes are composed purely of glacier ice, such that our flux divergence has a density of 900 kg m$^{-3}$. To determine the effective density of our elevation change signal, we consider the physical situation corresponding to the particular values of elevation change and flux divergence (Table S6). Where elevation change and flux divergence both have positive signs, we interpret mass accumulation as occurring and we assign a density of 600 kg m$^{-3}$. If both are negative, we assume this corresponds to ablation of glacier ice with a density of 900 kg m$^{-3}$. There is ambiguity about the state of glacier ice where flux divergence and elevation change are aligned, but this most likely corresponds to d$H$/d$t$ and SMB values close to zero, and we choose an intermediate density of 850 kg m$^{-3}$ to represent the variable likelihood of elevation change being composed of glacier ice or wetted snow and firn. We assume that the density uncertainty is approximately 60 kg m$^{-3}$ for all values[24].

**Uncertainty**. The uncertainty in the simplified continuity equation (Eq. 1) assuming independent errors is given analytically by

$$\sigma_{\dot{b}} = \sqrt{\left[\left(\frac{\sigma_{\nabla\mathbf{q}}}{\nabla\mathbf{q}}\right)^2 + \left(\frac{\sigma_{\rho_{\nabla\mathbf{q}}}}{\rho_{\nabla\mathbf{q}}}\right)^2\right]\left(\rho_{\nabla\mathbf{q}}\nabla*\mathbf{q}\right)^2 + \left[\left(\frac{\sigma_{dH}}{dH}\right)^2 + \left(\frac{\sigma_{\rho_{dH}}}{\rho_{dH}}\right)^2\right]\left(\rho_{dH}dH\right)^2}$$

$$(4)$$

In this equation, the flux divergence uncertainty for an individual pixel integrates the uncertainty for each of four fluxes dependent on multiple inputs (Eq. 2, Supplementary Fig. 1). Assuming these inputs to be subjected to completely random error would lead to an unrealistically high uncertainty estimate; given the ice thickness uncertainties, this would effectively assess the change in flux divergence due to a 40% change in ice thickness between adjacent pixels. Instead, we consider the uncertainty of each input dataset in terms of systematic bias and random error at the scale of an individual pixel and its neighbors. We assume that the uncertainties of the input datasets are not correlated to one another, and consider systematic and random errors separately for each.

We therefore derive the normalized ice thickness uncertainty $\frac{\sigma_h}{h}$ for each glacier as the standard error between individual ice thickness estimates on a pixel-by-pixel basis provided by[36], which we normalize relative to the consensus thickness estimate. We take the 68th centile value from the empirical distribution of normalized thickness standard errors (ie, 68% of standard errors are below this value; this is equivalent to the standard deviation for a one-sided distribution) as indicative of the glacier-wide systematic ice thickness uncertainty. We additionally consider that ice thickness is likely to have a random error component that the modeled ice thickness datasets do not reproduce, which we estimate to follow a gaussian distribution with ($\mu = 0$ m, $\sigma = 10$ m).

We use the pixel-wise ITS_LIVE reported error to derive the systematic normalized surface velocity uncertainty $\left|\frac{\sigma_u}{u}\right|$ for each glacier. Specifically, we use the reported error in surface velocity magnitude to determine the 68th centile value of $\left|\frac{\sigma_u}{u}\right|$ for each glacier, which we consider the systematic uncertainty. There is also a component of random error in the velocity data, but we assume that the random error is negligible at the scale of adjacent pixels in our analysis. We justify this assumption based on two factors. First, the velocity product is a synthesis of multiple years of observations and our target glaciers are non-surging mountain glaciers, which display consistent spatial patterns of velocity. We therefore expect that the flow direction and relative magnitude are generally very accurate, but that the multi-year mean speed is uncertain due to velocity change over the period of analysis and date biases in the data synthesis[16]; this is reflected by our systematic error. Second, we note that the $x$- and $y$-displacement uncertainty may be random at the scale of the velocity product (240 m) but is not likely to be random at the scale of adjacent pixels in our analysis. Our assumption is that pixel-scale patterns of ice velocity change accurately reflect larger-scale patterns of ice dynamics that are captured by the velocity data.

For the flux calculation, we assess the random error $\sigma_\gamma$ as the standard deviation of calculated $\gamma$ values from the 10,000 run Monte Carlo analyses for each glacier, described above. Considering the agreement in d$H$/d$t$ products from recent studies[15,20], we consider the uncertainty in d$H$/d$t$ to be limited to random error. We therefore use the reported uncertainty as $\sigma_{dH}$ in our Monte Carlo analysis. Finally, we assume the random uncertainty in density estimates to be 60 kg m$^{-3}$.

We integrate each source of uncertainty by perturbing our input data in a Monte Carlo analysis with 1000 distinct runs for each individual glacier. Using the uncertainties outlined above, we perturb our inputs with (1) random, spatially uncorrelated noise added to the d$H$/d$t$ data, (2) randomly chosen systematic scaling of the **u** data, (3) random, systematic scaling of the $h$ data, (4) random, uncorrelated noise added to the $h$ data, (5) random systematic scaling of the $\gamma$ estimates, (6) random variations in the density values $\rho_{dH}$ and $\rho_{\nabla\mathbf{q}}$. This enables us to estimate the integrated uncertainty in $\nabla*\mathbf{q}$ and $\dot{b}$. We also use the Monte Carlo results to determine the uncertainty in our derived metrics of ELA, AAR, committed volume loss, and balance ratio as the standard deviation of each metric for the full population of runs.

**Mass balance profiles**. Although our calculations are performed pixel by pixel across each glacier, slight inconsistencies between the observed velocity pattern and modeled ice thickness pattern can lead to an unrealistic pattern in flux divergence. This is due to several factors: (1) systematic decorrelation in the velocity product due to either a lack of identifiable features (particularly in accumulation areas) or rapid ice flow (particularly in icefalls), (2) the necessary use of a shape factor to distribute ice thicknesses across the glacier width[68,69] which can vary from glacier to glacier and even across glaciers, (3) the inability of current ice thickness models to treat glacier tributaries separately[36], and (4) the spatial variations of longitudinal stress gradients.

To mitigate this problem and to provide higher-confidence distributions of specific annual mass balance, we segment each glacier into hypsometric bins. To remove local undulations at the glacier surface, the ASTER GDEM v3[70] is resampled to the resolution of our analysis, then smoothed with an 11×11 Gaussian low-pass filter using a 2$\sigma$ threshold. We segment the resulting DEM into 25 m elevation bins, then intersect the result with a hole-filled version of the debris cover maps provided by[31]. For each segment, we determine the mean values of $\nabla*\mathbf{q}$, d$H$/d$t$, and $\dot{b}$, and the uncertainty is assessed for each variable through quadrature of

the distributed estimates. This aggregation step is crucial to reduce the effects of the factors listed above, and to resolve the overall pattern of SMB rather than amplifying noise due to errors in individual datasets. Finally, our SMB results are compared to available surface mass balance measurements from the World Glacier Monitoring Service[71] and other published literature, and to the results of[38] (Supplementary Table 1).

Based on the method's performance, we limit our analyses to larger glaciers (>2 km² in area) which are more likely to show a clear velocity signal[16]. We also remove surging glaciers from consideration for further processing, which we identify based on the RGI6.0 attributes. We additionally identify glaciers with erratic surface lowering or mass balance patterns, also indicative of surging or lower quality source data. In particular, we limit our glaciers for further analysis to those that satisfy the following conditions: the detrended altitudinal d$H$/d$t$ profile has a standard deviation of less than 3 m a$^{-1}$ and the d$H$/d$t$ profile has a nonnegative correlation with elevation. We consider these characteristics to be indicative of surging behavior. Finally, we only retain glaciers with the following criteria, which we consider to be indicative of higher quality input data and results: the optimized ELA has an Accuracy of at least 0.5, the detrended SMB profile has a standard deviation of less than 3 m w.e. a$^{-1}$; and the mean SMB uncertainty is less than 3 m w.e. a$^{-1}$. This leaves a population of 5527 glaciers representing 71% of the total ice volume of RGI regions 13, 14, and 15. Due to the quality controls, subregions are not uniformly sampled (Supplementary Tables 3, 4), which we account for (Regional Results).

**Determination of ELA and AAR**. The ELA is a single elevation contour ideally intended to distinguish between accumulation areas and ablation areas. Given our distributed mass balance dataset, we determine glacier-specific ELAs through an error minimization approach. We first classify pixels as accumulating or ablating mass based on the sign of SMB in our results. We then use each integer elevation within the glacier's elevation range as a binary classifier to produce a segmentation of accumulation and ablation areas. We assess each segmentation relative to our gridded results by determining the confusion matrix and computing its accuracy. We determine the ELA as the elevation that gives the best Dice coefficient[72] for the segmentation of accumulation and ablation areas (Supplementary Figs. 2–5). For glaciers whose optimal ELA is at either end of the glacier's elevation range (indicating mass loss or gain at all elevations), we fit a linear trend to the SMB and extrapolate significant trends to determine the elevation with SMB=0, which we take as an indicator of the theoretical climatic ELA for the glacier's location. For all glaciers for which we successfully resolve an ELA, we then determine the AAR as the portion of glacier area that lies above the ELA. We compare our ELA results to available datasets in the Supplementary Information.

**Calculation of ablation balance ratio**. Following[1], we calculate the balance portion of ablation (corresponding to the ratio of balance ablation to total ablation) for each glacier, subregion, and river basin. For each glacier, we calculate the total ablation directly based on our distributed SMB results, summing all pixels with a negative SMB. We then calculate the imbalance ablation for each glacier as the specific annual mass balance multiplied by glacier area[15]. From the total and imbalance ablation rates, we determine the rate of balance ablation (Supplementary Fig. 2). We express this for each glacier as a ratio of the balance ablation to total ablation, expressed as a percentage (Fig. 3); glaciers experiencing net annual accumulation thus have a balance ratio greater than 100%.

**Calculation of implied volume change**. To assess the volume change implied by our mass balance profiles, we developed a parameterization of glacier retreat and advance similar to[28,73]. In this framework, the annual mass balance is calculated based on our SMB results, and a △$h$ parameterization is used to redistribute this mass loss or gain across the glacier, updating the ice thickness of the glacier. The SMB dataset only changes based on glacier extent changes, eventually leading to an equilibrium state after numerous iterations. This parameterization approach has been demonstrated to appropriately represent glacier retreat by implicitly representing ice dynamics[73].

For each glacier with a clear signal of mass loss (mean mass balance less than −0.1 m w.e. a$^{-1}$), we develop a △$h$ parameterization based on the thinning rates from[15]. For glaciers with ambiguous thinning patterns, we use the △$h$ parameterization from[73] directly. For glaciers with a positive mass balance, we found that the 5 m a$^{-1}$ thickening threshold for advance used by[28] did not allow HMA glaciers to advance to a steady state. We therefore instead allow glaciers to advance when the terminus longitudinal gradient exceeds 10 degrees. We determine this longitudinal gradient based on the mean thickness of the lowest $N_t$ on-glacier pixels, where $N_t$ is the number of pixels equaling one glacier width in the terminus area, and the size of each pixel. This longitudinal gradient threshold was chosen such that the effective volume-area scaling relationship noted by[74] holds for our advancing glaciers. If a glacier is allowed to advance, the lowest-elevation $N_t$ glacier-marginal pixels become appended to the glacier, and are thickened by the prior terminus height, but the advancing fraction is limited to 50% of the glacier's total volume gain. The non-advancing mass accumulation is distributed altitudinally according to the original △$h$ parameterizations[28,73]. For advancing glaciers, we extrapolate the SMB from the glacier terminus at a rate of 0.07 m w.e. m$^{-1}$, which is the median observed ablation gradient in our extended database of field measurements (Supplementary Table 1).

We carry out this △$h$ parameterization for our subset of 5527 glaciers, updating ice thickness, glacier extent, SMB, and elevation datasets with an annual timestep. Although our simulations are performed for 200 years, we highlight the volume change results for the year 2100 as the △$h$ parameterization is most robust for multidecadal periods[73]. Finally, we note that these results depend strongly on the uncertainty of our SMB results, so we run 30 simulations for each glacier, varying the SMB systematically by the d$H$/d$t$ uncertainty, which is the primary source of uncertainty for glacier-wide mass balance. We therefore report the mean and standard deviation of regional glacier outcomes for 2100 implied by the current mass balance regimes and their uncertainty.

**Regional results**. We perform the above calculations for each glacier within our regional subset. We then aggregate these values to distinct subregions as defined by[14–16], and to major river basins[3] to provide a larger-scale perspective on the heterogeneity of glacier health. For ELA and AAR, we determine the area-weighted mean and its uncertainty, as well as the median value within each zone. For ablation balance and implied volume change, we aggregate total ice volumes but do not assume random uncertainty, instead determining the mean normalized uncertainty (Supplementary Tables 3, 4).

For the river basins we also seek to estimate total ablation including glaciers not represented in our regional subset due to their small size or low-quality input data. We therefore determine the total imbalance ablation in each basin from the results of[15] with our basin outlines and correcting for the subregional mass balance biases due to density estimates (Fig. 1). We then used the ratio of balance to imbalance ablation from our subset of glaciers for the basin to estimate the total ablation in the river basin. This leads to a different regional mean ablation balance ratio (60%) than for our subset (50%) by accounting for subregional sampling bias. As the uncertainty of the scaling is the major source of uncertainty for the regional balance ratios (Fig. 3), we do not scale the results of our future simulations, but report the regional aggregated values.

## Data availability
The SMB datasets generated and analyzed during the current study are available in the Zenodo repository, https://doi.org/10.5281/zenodo.3843292. The elevation change data of ref. [15] are available at https://doi.org/10.1594/PANGAEA.876545. The mean surface velocity data of ref. [16] are provided by the NASA MEaSUREs ITS_LIVE project and available at https://its-live.jpl.nasa.gov/. The consensus ice thickness dataset of ref. [36] is available at https://doi.org/10.3929/ethz-b-000315707. The glacier outlines of ref. [42] are available at https://www.glims.org/RGI/rgi60_dl.html. The supraglacial debris extents of ref. [31] are available at https://doi.org/10.5880/GFZ.3.3.2018.005. The WGMS Fluctuations of Glaciers database is available at https://doi.org/10.5904/wgms-fog-2019-12. River basin boundaries used in this study are available at http://www.fao.org/nr/water/aquamaps/. The Global Lakes and Wetlands Database is available at https://www.worldwildlife.org/pages/global-lakes-and-wetlands-database.

## Code availability
The code used to produce specific mass balance across High Mountain Asia and to derive implied volume loss is available on GitHub at https://github.com/miles916/HMA_continuity.

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

## Acknowledgements

This study would not have been possible without open data policies by key studies[15,16,36,71], each representing the work of a large body of scientists. We thank Mohd Farooq Azam and two anonymous reviewers whose comments and suggestions improved the study. This project has received funding from the European Research Council (ERC) under the European Union's Horizon 2020 research and innovation program grant agreement No 772751, RAVEN, "Rapid mass losses of debris-covered glaciers in High Mountain Asia". We acknowledge geospatial data provided by the USGS, NASA, and NASA/METI/AIST/Japan Spacesystems, as well as glaciological data provided by the NSIDC and WGMS.

## Author contributions

E.M., M.M., A.D., and F.P. designed the study and developed the methods. M.K. and S.F. supported with the collection of reference values of SMB and ELA. E.M. performed all calculations and led the writing of the paper, and all authors contributed to the interpretation of results and writing.

## Competing interests

The authors declare no competing interests.
