## [Peer Review File · Nature Communications]

REVIEWER COMMENTS

Reviewer #1 (Remarks to the Author):

Last summer, I reviewed the manuscript 'Health and Sustainability of Glaciers in High Mountain Asia' that Miles and colleagues submitted to 'Nature Geoscience'. At that time, I was overall positive about this submission, and stated that: "I have formulated a list of comments below. Most of these are not critiques, but mostly suggestions on how to improve the clarity of the manuscript. A few comments are slightly more substantial. I hope the authors find these comments useful, and I am convinced that most of them should be easily addressable, with little to no additional analyses to perform (except maybe for the committed ice loss experiments)".

I was pleased to see that the authors took into account my comments (including very elaborate and detailed rebuttal) and those by the other reviewers when reworking their manuscript (now under review for Nature communications). I particularly appreciated that the authors decided to re(de)fine how the committed mass loss/gain is calculated, by relying on the retreat parameterization by Huss et al. (2010). It is great to see that for this, the authors relied on observed thinning rates from Brun and colleagues (rather than on the more widely used 'rough' thinning rates that were initially derived from Alpine glaciers), and that a special methodology was developed for advancing glaciers. These changes involved a substantial amount of additional work, but this has definitely increased my confidence in the numbers put forward for this section.

At this stage, I have no new or further comments. I'd be glad to endorse the manuscript in its current shape, and I am convinced that it could become an important study in our field of research.

Reviewer #2

Review: Miles et al.

The manuscript is clear, nicely written and brings substantial advancement in HMA glaciology. The methods are well established but applied first time at this large scale (applications only on individual glaciers were available). Uncertainty estimation is done carefully. I don't find any flaw in methods and uncertainty estimation. Results will be of great importance for regional or even glacier-scale model validation. In my opinion, the manuscript has good potential for Nat. Com.

I have already reviewed the manuscript for Nat. Geo. and provided the detailed comments. Authors have replied satisfactorily to all of my comments. However, I still have a few comments on the revised manuscript which need to be addressed.

Validation of method and selection of glaciers for this study (L 69-78): Authors have validated their method using field-observed data on 35 glaciers, almost half of them are having area less than 2 km² (a few of them are even less 0.5 km²), and then applied their method on 5527 glaciers having an area of 2 km² or greater. A few reasons for not including small glaciers (< 2 km²) are given in supplementary. I feel if the method is able to reproduce 76% (by the way 79% in supp. mat.) of field measurements within 0.2 m w.e./yr then why not apply the method on all glaciers? It may be computationally expensive though. Perhaps authors should also check the validation on glaciers having an area > 2 km², it might give even better validation as small glaciers are not dynamically very active. If authors decide to exclude the glaciers with an area <2 km² then limitation of method should be given clearly in the main manuscript.

Sustainable ablation in major basins: This is very interesting section and a significant advancement to the ablation (imbalanced) contribution to river runoff, done using geodetic mass balances (Kääb et al., 2012; Brun et al., 2017 etc.). However, I think authors need to be more careful here and provide clearer information:

(1) Idea for 'balance ablation' is said to be rooted to Pritchard (2019), in which a method (based on mean monthly precipitation and positive temperatures) of Kesar et al. (2010) was used. While in the present manuscript, authors have estimated the 'balance ablation' from accumulation that have estimated with their method (Figure S2).

(2) This method cannot be applied if the glacier mass balance (or regional mass balance) is positive when some of the accumulation is actually stored for next years on the glacier. For instance, glacier mass balances (balanced or slightly positive) in Kunlun Shan and Karakoram cannot be resolved in 'balance ablation' and 'imbalance ablation'. Perhaps this is the reason, authors decided to estimate the basin-wide 'balance ablation' and 'imbalance ablation'. For instance, in Indus Basin, the Karakoram glaciers are in balance while other regions of Indus (Lahaul-Spiti, Hindu-Kush) are imbalanced, providing the negative basin-wide mass balances. This limitation should be explained in the main manuscript.

L 119-121: Is it because you have excluded the surging glaciers from your selected population?

Figure: 3: Basin vulnerability of Indus and Ganges-Brahmaputra is same in this figure while Figure S21 shoes otherwise.

Reviewer #1 (Remarks to the Author):

Last summer, I reviewed the manuscript 'Health and Sustainability of Glaciers in High Mountain Asia' that Miles and colleagues submitted to *Nature Geoscience*. At that time, I was overall positive about this submission, and stated that: 'I have formulated a list of comments below. Most of these are not critiques, but mostly suggestions on how to improve the clarity of the manuscript. A few comments are slightly more substantial. I hope the authors find these comments useful, and I am convinced that most of them should be easily addressable, with little to no additional analyses to perform (except maybe for the committed ice loss experiments)'.

I was pleased to see that the authors took into account my comments (including very elaborate and detailed rebuttal) and those by the other reviewers when reworking their manuscript (now under review for *Nature* communications). I particularly appreciated that the authors decided to re(de)fine how the committed mass loss/gain is calculated, by relying on the retreat parameterization by Huss et al. (2010). It is great to see that for this, the authors relied on observed thinning rates from Brun and colleagues (rather than on the more widely used 'rough' thinning rates that were initially derived from Alpine glaciers), and that a special methodology was developed for advancing glaciers. These changes involved a substantial amount of additional work, but this has definitely increased my confidence in the numbers put forward for this section.

At this stage, I have no new or further comments. I'd be glad to endorse the manuscript in its current shape, and I am convinced that it could become an important study in our field of research.

We thank the reviewer for their careful consideration of our work in both iterations, as well as the helpful suggestions. We are very happy that they appreciate the effort put into the revision following all reviewers' suggestions, in particular the substantial changes to the implied mass change calculations. Without a doubt, the study has substantially benefited from the reviewer's earlier comments and suggestions.

Reviewer #2 (Remarks to the Author):

Review: Miles et al.

The manuscript is clear, nicely written and brings substantial advancement in HMA glaciology. The methods are well established but applied first time at this large scale (applications only on individual glaciers were available). Uncertainty estimation is done carefully. I don't find any flaw in methods and uncertainty estimation. Results will be of great importance for regional or even glacier-scale model validation. In my opinion, the manuscript has good potential for Nat. Com.

I have already reviewed the manuscript for Nat. Geo. and provided the detailed comments. Authors have replied satisfactorily to all of my comments. However, I still have a few comments on the revised manuscript which need to be addressed.

We thank the reviewer for their careful consideration of our work in both iterations, which have significantly helped to shape and improve the study. We provide responses to the additional comments below.

Validation of method and selection of glaciers for this study (L 69-78): Authors have validated their method using field-observed data on 35 glaciers, almost half of them are having area less than 2 km² (a few of them are even less 0.5 km²), and then applied their method on 5527 glaciers having an area of 2 km² or greater. A few reasons for not including small glaciers (< 2 km²) are given in supplementary. I feel if the method is able to reproduce 76% (by the way 79% in supp. mat.) of field measurements within 0.2 m w.e./yr then why not apply the method on all glaciers? It may be computationally expensive though. Perhaps authors should also check the validation on glaciers having an area > 2 km², it might give even better validation as small glaciers are not dynamically very active. If authors decide to exclude the glaciers with an area <2 km² then limitation of method should be given clearly in the main manuscript.

This is an interesting question. We had chosen not to analyze the smaller glaciers primarily because of the frequency with which no velocity was observed for these glaciers (as reported by ITS_LIVE), which is shown as qualitative notes in Table S1 for most of the 11 smaller glaciers in our validation dataset. Following the reviewer's suggestion, we separately evaluated the method's performance for glaciers smaller and larger than 2 km². Interestingly, we find little difference between the performance for small and large glaciers (81% and 78% of observations within 0.2 m w.e./a of our results, respectively), despite the absence of a flux divergence correction for most small glaciers. This implies that there is little flux divergence for many of these glaciers, at least in the region of the stake measurements (recalling that few glaciers have accumulation measurements in this region).

We could therefore apply our method to all glaciers in High Mountain Asia without any apparent loss in accuracy of the SMB product. However, without measurable velocity, which our method relies on to derive mass accumulation, the result of our method would be the Brun et al (2017) thinning pattern with a different density assumption, which would not be meaningful or correct for these glaciers, and which would affect the balance ablation and committed loss estimates at the regional and basin scales.

To elaborate further, the challenge is that no observation of glacier motion does not necessarily correspond to no glacier motion. In the figure below, we show the effect of our corrections for small and large glaciers at the stake locations, highlighting that the small glaciers' results appear to be accurate because there is no apparent flux divergence. There is considerable flux divergence for larger glaciers,

however, and our method is able to reproduce this for most stakes. At some stake locations the method fails due to velocity non-observation.

Figure R1. Here we depict on the left the disagreement between stake measurements and the dH/dt from Brun et al (2017) on the x-axis, which corresponds to the effective flux divergence at each stake. On the y-axis is the disagreement between the stake measurements and our SMB results. From this it is clear that small glaciers' stake observations are clustered around the origin and the 1:1 line, indicating no flux divergence correction. The larger glaciers are also clustered around the origin (flux divergence = 0) but are clustered along the y=0 line (perfect flux divergence correction) for a range of apparent flux divergence values, while it is not currently possible to produce robust SMB estimated at many stakes with our method due to velocity data inaccuracies. These are generally removed from our results (thus $N=5527$ instead of 8099). The right panel clearly shows that the locations with high error were associated with low rates of observed surface velocity.

Consequently, the reviewer's suggestion is an excellent stimulus for future study. In fact, we are keen to use more recent, highly-resolved velocity products (such as that of Millan et al., 2019) to determine the mass balance for small glaciers, but such datasets are currently only available for the recent period (2016-present). Assembling dH/dt and velocity measurements for this recent period will be the scope of a subsequent study that will focus on smaller glaciers.

Taking these factors into account, we have indicated more clearly in the main manuscript where the method fails and justified the limitation to glaciers larger than 2 km², for which velocity products are more reliable at present.

Line 72-73: 'Our method is consistent with 79% of field measurements to within 0.2 m w.e. a⁻¹ and generally reproduces observed mass balance patterns *where glacier velocity is measurable* (Supp Mat).'

Line 74-75: 'We thus apply this method to the 7341 glaciers in Central and Southern Asia with all necessary inputs and an area of 2 km² or greater, *for which velocity is generally resolved well.*'

Line 247: 'Problems in the input datasets of velocity and ice thickness forced us to discard results for 25% of the 7341 glaciers analysed, *and prevented application to glaciers smaller than 2 km².*'

Thank you, by the way, for catching the typo, which has been corrected in the main text. The 79% value was correct.

Sustainable ablation in major basins: This is very interesting section and a significant advancement to the ablation (imbalanced) contribution to river runoff, done using geodetic mass balances (Kääb et al., 2012; Brun et al., 2017 etc.). However, I think authors need to be more careful here and provide clearer information: (1) Idea for 'balance ablation' is said to be rooted to Pritchard (2019), in which a method (based on mean monthly precipitation and positive temperatures) of Kesar et al. (2010) was used. While in the present manuscript, authors have estimated the 'balance ablation' from accumulation that have estimated with their method (Figure S2). (2) This method cannot be applied if the glacier mass balance (or regional mass balance) is positive when some of the accumulation is actually stored for next years on the glacier. For instance, glacier mass balances (balanced or slightly positive) in Kunlun Shan and Karakoram cannot be resolved in 'balance ablation' and 'imbalance ablation'. Perhaps this is the reason, authors decided to estimate the basin-wide 'balance ablation' and 'imbalance ablation'. For instance, in Indus Basin, the Karakoram glaciers are in balance while other regions of Indus (Lahaul-Spiti, Hindu-Kush) are imbalanced, providing the negative basin-wide mass balances. This limitation should be explained in the main manuscript.

This is also an excellent comment. We absolutely agree with the reviewer's point (1) that it is worth noting that the Pritchard (2019) and Kaser et al (2010) studies used modelled precipitation rates for their calculation of balanced ablation, whereas we instead use the annual accumulation derived from the continuity equation. In this respect, we do think that our results represent an improvement over the past balance/imbalance estimates as not all annual precipitation leads to accumulation, but more importantly, our analysis allows us to isolate the value for individual glaciers, rather than at the catchment or basin level. This is indeed an important distinction, which we have highlighted in the revised manuscript (lines 152-153):

'Crucially, the SMB results allow us to determine these values directly for each glacier, whereas prior available estimates were obtained *only at the basin scale* by comparing observed thinning with glacier models^{1,20}'

However, we believe that the reviewer has a slight misconception with regards to point (2), possibly due to the color range in Figure 3. First, we do resolve the balance ablation for individual glaciers (see the small points in Figure 3) as well as for the river basins. In fact, we are confident that the balance ablation calculation does provide meaning for glaciers that are growing over our 16-year study period; in this case the balance ablation (which is equivalent to the total annual accumulation) is greater than the total annual ablation, as mentioned in the Methods section (former manuscript line 479): '*glaciers experiencing net annual accumulation thus have a balance ratio greater than 100%.*' This actually does occur for the majority of glaciers in the Kunlun Shan and W Karakoram, as well as many glaciers in the Pamirs, but we had limited the color scale to 100%. This was done only because a glacier with 100% balance ablation over a period is 'perfectly' in balance, but may have led to this misunderstanding. For clarity in the revised version of the manuscript, we have therefore extended this range to 120% with additional color elements to depict the degree to which growing glaciers have accumulation in excess of their ablation, which are primarily located in the Kunlun Shan. We have indicated this color scale clearly in the revised Figure caption:

'Analyzed glaciers are colored according to the portion of total annual ablation that is compensated by accumulation, *which is greater than 100% for glaciers gaining mass.*'

We have also modified the main text to point to this directly (lines 163):

'Nonetheless, we show that these basins' glaciers were much healthier compared to the rest of HMA for our study period, with over 50% of annual glacier ablation balanced by accumulation *and numerous individual glaciers exceeding 100% balanced ablation* (Figure 3).'

For convenience, the modified Figure 3 appears here:

Figure 3. Quantity and context of glacier melt for principal High Asian drainage basins. All individual aAnalyzed glaciers are colored according to the portion of total annual ablation that is compensated by accumulation, which is greater than 100% for glaciers gaining mass. The portion of balance ablation derived from our results is shown also for major river basins, indicating uncertainty with the dashed lines, and scaled by area according to the total estimated glacier ablation within each (Methods, Tables S4 & S5). Basin vulnerability is colored according to the global range³. Background is a hillshade of the GTOPO30 dataset sourced from the USGS (<https://doi.org/10.5066/F7DF6PQS>).

L 119-121: Is it because you have excluded the surging glaciers from your selected population?

This is a good question relating to the continuous variations of ELA around and through the Karakoram and Kunlun Shan. Without detailed observation of the surging glaciers themselves we cannot say for certain, but we do not believe that the surging glaciers have substantially different equilibrium line altitudes to the non-surging glaciers surrounding them. Surging glaciers in this region tend to be larger (Sevestre and Benn, 2015; Goerlich, Bolch, and Paul, 2020) and although surging glaciers are characterized by distinct elevation change profiles during quiescent and advance stages (e.g. Round et al., 2017), surging and advancing glaciers typically do not exhibit markedly different glacier-wide mass balances (e.g. Lv et al, 2020) and show similar dynamic response to thinning (Dehecq et al., 2019). This makes sense, as surging is the ice flow manifestation of an oscillation in the combined mass and enthalpy budgets (Benn et al., 2019). We see little reason to believe that their mass inputs or altitudinal melt rates would be strongly distinct.

An alternative possibility indicated by recent research is that in recent decades this area has seen an increase in precipitation due to large-scale groundwater extraction and flood irrigation (de Kok, 2018; 2020). This process could certainly lead to the increase in AAR for this area without a clear ELA pattern. We think that the topographic availability explanation is equally compelling and complementary to this hypothesis of a process by which the excess accumulation is provided. We have altered the description in the text at line 126-127 to include this hypothesis:

'Consequently, recent increases in high-altitude precipitation⁵¹ would affect a disproportionately large portion of glacier area in this subregion.

Figure: 3: Basin vulnerability of Indus and Ganges-Brahmaputra is same in this figure while Figure S21 shows otherwise.

Thank you for this question. Indeed the vulnerability values for the Indus and Ganges-Brahmaputra Basins differ slightly (0.55 and 0.53, respectively). We have checked and the correct color was coded in Figure 3 for this value, although the contrast is difficult to perceive. In fact, we do not expect that this difference in value is particularly meaningful, as both the Indus and Ganges-Brahmaputra Basins are very vulnerable (0.53-0.55) relative to the global range of values from Immerzeel et al (2020). These basins rank as the 6th and 7th most vulnerable basins globally. Consequently we have not made a change to the coloring of the Figure.

Benn, D., Fowler, A., Hewitt, I., & Sevestre, H. (2019). A general theory of glacier surges. *Journal of Glaciology*, 65(253), 701-716. doi:10.1017/jog.2019.62

Brun, F., Berthier, E., Wagnon, P., Käab, A., & Treichler, D. (2017). A spatially resolved estimate of High Mountain Asia glacier mass balances from 2000 to 2016. *Nature Geoscience*, (10), 668–673. <https://doi.org/10.1038/ngeo2999>

Dehecq, A., Gourmelen, N., Gardner, A. S., Brun, F., Goldberg, D., Nienow, P. W., ... Trouvé, E. (2019). Twenty-first century glacier slowdown driven by mass loss in High Mountain Asia. *Nature Geoscience*, 12(1), 22–27. <https://doi.org/10.1038/s41561-018-0271-9>

de Kok, R. J., Tuinenburg, O. A., Bonekamp, P. N. J., & Immerzeel, W. W. (2018). Irrigation as a Potential Driver for Anomalous Glacier Behavior in High Mountain Asia. *Geophysical Research Letters*, 45(4), 2047–2054. <https://doi.org/10.1002/2017GL076158>

de Kok, R. J., Kraaijenbrink, P. D. A., Tuinenburg, O. A., Bonekamp, P. N. J., & Immerzeel, W. W. (2020). Towards understanding the pattern of glacier mass balances in High Mountain Asia using regional climatic modelling. *The Cryosphere*, 14(9), 3215–3234. <https://doi.org/10.5194/tc-14-3215-2020>

Goerlich, F., Bolch, T., & Paul, F. (2020). More dynamic than expected: An updated survey of surging glaciers in the Pamir. *Earth System Science Data Discussions*, (April), 1–26.

Immerzeel, W. W., Lutz, A. F., Andrade, M., Bahl, A., Biemans, H., Bolch, T., ... Baillie, J. E. M. (2020). Importance and vulnerability of the world's water towers. *Nature*, 577(7790), 364–369. <https://doi.org/10.1038/s41586-019-1822-y>

Kaser, G., Großhauser, M., Marzeion, B., & Barry, R. G. (2010). Contribution potential of glaciers to water availability in different climate regimes. *Proceedings of the National Academy of Sciences of the United States of America*, 107(47), 21300–21305. <https://doi.org/10.1073/pnas>.

Lv, M., Guo, H., Lu, X., Liu, G., Yan, S., Ruan, Z., ... Quincey, D. J. (2019). Characterizing the behaviour of surge- and non-surge-type glaciers in the Kingata Mountains, eastern Pamir, from 1999 to 2016. *Cryosphere*, 13(1), 219–236. <https://doi.org/10.5194/tc-13-219-2019>

Millan, R., Mouginot, J., Rabatel, A., Jeong, S., Cusicanqui, D., Derkacheva, A., & Chekki, M. (2019). Mapping surface flow velocity of glaciers at regional scale using a multiple sensors approach. *Remote Sensing*, 11(21), 1–21. <https://doi.org/10.3390/rs11212498>

Pritchard, H. D. (2019). Asia's shrinking glaciers protect large populations from drought stress. *Nature*, 569(7758), 649–654. <https://doi.org/10.1038/s41586-019-1240-1>

Round, V., Leinss, S., Huss, M., Haemmig, C., & Hajnsek, I. (2017). Surge dynamics and lake outbursts of Kyagar Glacier, Karakoram. *Cryosphere*, 11(2), 723–739. <https://doi.org/10.5194/tc-11-723-2017>

Sevestre, H., & Benn, D. I. (2015). Climatic and geometric controls on the global distribution of surge-type glaciers: Implications for a unifying model of surging. *Journal of Glaciology*, 61(228), 646–662. <https://doi.org/10.3189/2015JoG14J136>

REVIEWERS' COMMENTS

Reviewer #2 (Remarks to the Author):

Authors have done excellent job and included all of my concerns satisfactorily. I don't have any further comment on the manuscript and endorse it for publication in Nat. Comm.

Mohd. Farooq Azam

REVIEWERS' COMMENTS

Reviewer #2 (Remarks to the Author):

Authors have done excellent job and included all of my concerns satisfactorily. I don't have any further comment on the manuscript and endorse it for publication in Nat. Comm.

Mohd. Farooq Azam

We are delighted that the reviewer is satisfied with our revisions according to their thoughtful comments, and we would like to thank the reviewer for their very constructive comments, which helped to refine the study.